# Hand-Arm Vibration Assessment and Changes in the Thermal Map of the Skin in Tennis Athletes during the Service

**DOI:** 10.3390/ijerph16245117

**Published:** 2019-12-14

**Authors:** Ana M. Amaro, Maria F. Paulino, Maria A. Neto, Luis Roseiro

**Affiliations:** 1CEMMPRE, Department of Mechanical Engineering, University of Coimbra, 3004-531 Coimbra, Portugal; maria.paulino@dem.uc.pt (M.F.P.); augusta.neto@dem.uc.pt (M.A.N.); lroseiro@isec.pt (L.R.); 2Coimbra Polytechnic-ISEC, 3030-199 Coimbra, Portugal

**Keywords:** hand-arm vibration, infrared thermography, over-grip, tennis service

## Abstract

During recent years the number of tennis athletes has increased significantly. When playing tennis, the human body is exposed to many situations which can lead to human injuries, such as the so-called tennis elbow (lateral epicondylitis). In this work a biomechanical analysis of tennis athletes, particularly during the service, was performed, considering three different types of over-grip and the presence of one anti-vibrator device. One part of the study evaluates the exposure to hand-arm vibration of the athlete, based on the European Directive 2002/44/EC concerning the minimum health and safety requirements, regarding the exposure of workers to risks from physical agents. The second part of the study considers an infrared thermography analysis in order to identify signs of risk of injury, particularly tennis elbow, one of the most common injuries in this sport. The results show that the presence of the anti-vibrator influences the vibration values greatly in the case of athletes with more experience and also for athletes with less performance. The presence of the Cork and/or Tourna on the racket grip does not have any significant effect on the hand-arm vibration (HAV), similarly in the case of athletes with the best performance and athletes with less technique. The results indicated that the infrared thermography technique may be used to identify the risk of injuries in tennis players.

## 1. Introduction

There is a significant number of tennis players in the world, taking into consideration both the professionals and those who play for leisure. Tennis can be played on a variety of surfaces, such as a hard court (acrylic), clay, grass, and artificial grass. As it is a sport that is played at a high intensity there is a great propensity for injuries. A tennis athlete should have excellent physical preparation, because there is no fixed time to finish the game. Tennis athletes are susceptible to a range of injuries including conditions caused by chronic overuse and acute traumatic injuries [1]. Disorders of the musculoskeletal system can appear, such as tennis elbow, shoulder injuries, stress fractures, and muscle strains, which cause health problems [2]. Ferrara and Cohen [3] wrote that the repetitive impact and abuse of the upper extremities in racquet sports can increase the risk of tissue fatigue and injury, because during the impact between the racket and the ball the muscles are used the most. According to Chadefaux et al. [4], the racket has an important role in the athlete’s performance, and also in the prevention of pathologies and discomfort. Hennig [5] did a study to evaluate the influence of the racket on injuries in tennis athletes, and concluded that one of the causes for developing tennis elbow is the number of vibrations transmitted to the arm by the hand, due to the impact between the racket and the ball. Baszczyński et al. [6] evaluated the spring tension and mass of a tennis racket and determined that higher spring tension causes more vibration, but if a mass was added to the racket the amplitude of vibrations is reduced, because this mass acts as a shock absorber that helps to dissipate kinetic energy. According to many authors, the most common injury observed in tennis athletes occurs in the elbow. Kachanathu et al. [7] saw that the most acute injuries occurred in the body joints, such as in the elbows, followed by wrists, ankles, shoulders, and knees. They also observed that lateral epicondylitis (tennis elbow) occurs more frequently in recreational rather than in professional tennis athletes. These injuries may also be associated to the chronic repetitive overload. Chung and Lark [8] say that an incorrect position of the tennis athlete and his or her level of expertise could have a significant influence on the risk of the occurrence of injuries. Normally the elbow joint is stabilized, but the kinetic chain of the tennis service changes this, and promotes variations, for example, in the feet, knees, shoulder, elbow joint, and hand [9]. Kovacs and Ellenbecker [10] did a study using an 8-stage model to evaluate the tennis service. For them, this component of tennis is the most complex stroke among competitive athletes. According to Williams and Hebron [11] there is no substantial difference between service and groundstrokes in relation to shoulder injuries. However, they say that the upper limbs have been found to account for between 20% and 49% of injuries, with the shoulder and elbow being most frequently injured. During the tennis service there are five phases to be considered: wind-up (knee flexion, trunk rotation), early cocking, late cocking (position of maximal abduction-external rotation), acceleration phase (including long axis rotation), and follow through, which can promote, due to the repetition, an increased risk of overloading various structures around the shoulder [12]. Allen et al. [13] say that due to the great consistency of the impact conditions, and the advantage that the athlete has in developing a high-speed overhead service, the biomechanical consistency in serving is higher than the other strokes.

Welcome et al. [14] say that the hand-arm vibration (HAV) exposure is directly associated to the hand-arm vibration syndrome (HAVS), which is a syndrome that affects the blood vessels, joints, muscles, and nerves of the hand, wrist, and arm. These authors have concluded that the vibration transfer depends on the vibration direction and the measurement location. The HAVS may be developed due to a prolonged and intensive exposure to vibration [15]. According to López-Alonso et al. [16] the signs of exposure to HAV may be classified as vascular, neurological, or musculoskeletal. Fridén [17] says that the exposure to HAV can cause a variety of vascular and neuromuscular symptoms, like tingling in the fingers, discomfort or inflammation in the wrist and hand, digital lightening, intolerance to cold, feebleness of the finger flexors or basic muscles, and discoloration and trophic skin lesions of the fingers. One of the most common injuries in tennis athletes, as mentioned before, is tennis elbow, which can be described as a specific pathological disorder originating from the musculature of the lateral epicondyle, generally caused by overload [18]. An incorrect technique during the service may be the factor that most frequently causes tennis elbow, which can be reduced if the tennis athlete manages to perform the service using both hands alternatively and not only the predominant hand [19]. Nevertheless, this seems to be difficult due to the technique employed by tennis athletes during the service.

Nowadays, there are a relatively great number of techniques and studies that include motion capture/analysis, vibration analysis, measurement of human performance, etc. [20,21,22,23], but there are few studies using infrared thermography (IRT). Nevertheless, the use of the IRT technique to evaluate skin temperature in health and sports is constantly growing, which shows the importance of this technique concerning performance in sports [24,25,26,27,28,29,30]. IRT makes detection of the heat pattern and blood flow possible by the use of a special camera to measure the temperature of the skin and is a safe and a noninvasive technique. The use of IRT in sports is possible due to the fact that the body temperature is one of the principal indicators used to analyze the health status of a human, which can change during and after the practice of sports. IRT is also used to evaluate the thermal body patterns of healthy adults, in order to understand whether significantly higher differences between the contralateral skin temperatures could indicate serious problems, like hyperthermia or hypothermia, or others [31]. For some authors the results obtained by using the IRT can provide fundamental information for athletes, sports physiotherapists, coaches, and dieticians. IRT is used to evaluate musculoskeletal disorders during the practice of sports, and can help to prevent injuries, by identifying warning signs before the damage occurs, because the temperature of the affected areas, when compared to other similar ones, normally changes in the case of overuse [32]. del Estal et al. [33] use the IRT to evaluate the thermal asymmetries in striking combat athletes, like kickboxing and Muay Thai athletes. These authors selected 18 regions of interest (ROIs) and the asymmetries were analyzed by comparing lead and rear sides. According to Uchôa et al. [34] the variations observed in the skin surface temperature, using IRT analysis, detected after resistance training may be considered an indicator that the muscles could be inflamed. These authors have confirmed that the skin temperature seems to be an important variable to be considered in sports and can help to prevent injuries in the practice of them.

There are a lot of studies concerning the injuries in tennis athletes. However, according to these authors’ knowledge, no studies have been done to evaluate the HAV, simultaneously with the IRT, to identify the probability of injury in tennis athletes during the service, until now. The aim of this study is to evaluate the HAV and changes in body skin temperature in tennis athletes during the service. The authors intend to contribute to help tennis athletes reduce the risk of injury with their results by identifying the signs of injuries before they occur.

## 2. Materials and Methods

All participants gave written informed consent prior to participation, according to the Helsinki declaration. This research study applied the calculation method outlined in the International Standard [35,36]. Six male volunteers, all tennis athletes, with a mean age of 21 years and an experience of over 10 years, which places them in an elite group, participated in this study (Table 1).

Only one racket, Wilson Pro Staff 97, and one ball type, Wilson US OPEN number 4, were used by all the athletes in all tests. The racket was used under four different conditions, defined by the type of grip used and the use or not of an anti-vibrator as shown in Figure 1: the over-grip Wilson Pro (with a thickness of 0.6 mm, presents a great traction, which allows a reduction in sliding) and without an anti-vibrator (WGNAV); the over-grip Wilson Pro with an anti-vibrator (WGWAV); the over-grip Tourna (which is considered superfine, with a thickness of 0.45 mm and is normally used when needing to absorb more sweat) and with an anti-vibrator (TGWAV) and an over-grip developed in cork (a combination, made by the authors, combining the Wilson Pro over-grip and 2 mm of cork, with a total thickness of 2.6 mm) with an anti-vibrator (CGWAV). All the volunteers performed at least 10 services under each test condition. Table 2 presents the randomized order used in this study.

### 2.1. Hand-Arm Vibration (HAV)

In order to quantify the HAV induced in the athletes, one triaxial accelerometer (ICPR/IEPE Integrated Circuit Piezoelectric) was placed over the grip, at a distance of 10.9 cm from the tip of the over-grip (Figure 2). The coordinate system used to acquire HAV data is defined in Figure 3. The accelerometer channels were aligned with the racket, in such a way that x is the anterior–posterior direction, y is the media–lateral direction, and z is the superior–inferior direction.

The data acquisition system includes an NI 9234 four channel simultaneous sampling IEPE module (National Instruments, Austin, TX, USA) and a wireless NI WLS-9234 chassis connected to a laptop computer equipped with LabVIEW^®^ software (National Instruments: Austin, TX, USA). The WLS-9234 delivers 102 dB of dynamic range and incorporates software-selectable AC/DC coupling and IEPE signal conditioning for accelerometers and microphones. The four input channels make it possible to digitize simultaneous signals at rates of up to 51.2 kHz per channel and permit built-in antialiasing filters that automatically adjust to the sampling rate. As for the HAV, the frequency range recommended by ISO 5349 is 6.3–1000 Hz and the sampling frequency of the HAV was set at 1000 Hz. The acceleration data was filtered, as specified in the standards, using the SVT Human Vibration tool of LabVIEW (LabView, Manual of NI Sound and Vibration Measurement Suite 6.0, 2007). The filter coefficients were evaluated according to the procedures developed by Rimmel and Mansfield [37].

In order to quantify the vibration of exposure the authors used the European Directive 2002/44/EC, which defines the following terms for workers related to the HAV:

Hand-arm vibration: when mechanical vibration is transmitted to the hand-arm some risks to the health and safety of the workers appear, like vascular disorders, osteoarticular injuries, and neurological or muscular disorders.

Daily exposure action value (DEA): if the vibration exposure value exceeds 2.5 ms^−2^ in an 8-h reference period for a vibration dose value, the implementation of a program of technical and/or organizational measures is necessary to reduce the exposure to the mechanical vibration.

Daily exposure limit value (DEL): the vibration exposure value cannot, under any circumstances, exceed the value of 5 ms^−2^ in an 8-h reference period.

The daily exposure value for an 8-h reference period, A(8), for the HAV, is defined by:(1)$$A_{i}\left(8\right)=a_{w_{i}}\sqrt{\frac{T_{d}}{T_{0}}};\;\left(i=1,2,3\right),$$
where the index *i* is used to indicate either the *x*, *y* or *z* axes, respectively, $a_{w_{i}}$ is the weighted root means square acceleration, rms, specified in (ms^−2^) in the direction of the three axes, $T_{d}$ is the daily duration of exposure to the vibration $a_{w_{i}}$, and $T_{0}$ is the 8-h reference period (28,800 s).

The rms value of the acceleration over a direction *i* is defined as
(2)$$a_{w_{i}}={\left[\frac{1}{T}\int_{0}^{T}a_{w_{i}}^{2}\left(t\right)\;dt\right]}^{1/2};\;\left(i=1,2,3\right),$$
where *T* is the duration of the measurement and $a_{w_{i}}(t)$ is the weighted acceleration specified in (ms^−2^) in the direction of the corresponding three axes. The ISO 5349 standards specify that the total magnitude of the vibration should be determined from the vibration in the (x,y,z) orthogonal coordinates, in the case of HAV, as
(3)$$a_{v}={\left(a_{w_{1}}^{2}+a_{w_{2}}^{2}+a_{w_{3}}^{2}\right)}^{1/2}.$$

It is possible to obtain a relation between the parameter value A(8) and the number of years of exposure, *D*:(4)$$D=31.8\;{\left[A\left(8\right)\right]}^{-1.06},$$
which enables the vibration exposure limit until there is 10% probability of an individual developing vibration white finger (also denominated Raynaud’s disease).

### 2.2. Infrared Thermography (IRT)

The IRT was performed using a thermographic camera T430sc (FLIR^®^ Systems, Täby, Sweden) 320 × 240 pixels of resolution, spectral range of 7.5–13 μm, image frequency of 60 Hz, accuracy of ±2 °C, and sensitivity <0.030 °C (NETD—Noise Equivalent Temperature Difference). The pictures were taken of the volunteers’ naked trunk. The images were analyzed using the software FLIR ResearchIR Max. Using this software, it was possible to evaluate the maximum, mean, and minimum temperatures over the several ROIs. A total of 20 ROIs (Figure 4) were analyzed: right biceps (RB), left biceps (LB), right sternocleidomastoid (RS), left sternocleidomastoid (LS), right oblique (RO), left oblique (LO), right pectoral (RP), left pectoral (LP), serratus anterior right (SAR), serratus anterior left (SAL), right deltoid (RD), left deltoid (LD), lumbar (L), trapezius (T), right deltoid lateral perspective (RDR), left deltoid lateral perspective (LDL), right sternocleidomastoid lateral perspective (RSR), left sternocleidomastoid lateral perspective (LSL), right extensor carpi radialis brevis (RECRB), and left extensor carpi radialis brevis (LECRB). Descriptive time intervals of 0.4 s (*T_sk_*) and the correspondent standard deviation (St.Desv.). Thermal skin variations $\Delta T_{sk}$ were calculated as the difference between the average skin temperature value after the 40 services $\left(T_{sk,40}\right)$ and the average skin temperature value at the reference situation, before performing any service, $\left(T_{sk,0}\right)$, according to Equation (5):(5)$$\Delta T_{sk}=T_{sk,40}-T_{sk,0}.$$

All the volunteers were submitted to the same protocol to carry out the tests (Table 3). The racket used was the Wilson Pro with anti-vibrator. The thermographic camera was placed at a distance of 3 m from each volunteer. Initially, each athlete remained seated in a chair in a comfortable position for a period of 8 min so that the body temperature stabilized. After this several thermographic photographs were taken from four different perspectives: front, rear, right side, and left side. These photographs allowed to obtain the thermal map before the exercise execution, thus corresponding to the athlete’s reference standard. Then the player in the test performed 10 services within 1 min. Next, and during 1 min, the player was again photographed with the camera in the same position. Immediately thereafter, the player repositioned himself to serve another 10 services and was rephotographed again for a period of 1 min and in the same sequence of image acquisition. This procedure was repeated to make a total of 40 services and 20 photos per player. It should be noted that all photos were taken of the bare-chested player in order to be able to assess the skin temperature in the desired areas, due to the fact that the thermal imager can only obtain the skin surface temperatures. A room temperature of 18 °C was measured and kept constant during all the tests. For each volunteer and over-grip considered a total of 40 services and 20 photos were realized, making a total of 960 evaluations and 480 photos.

In order to statistically evaluate the influence of the conditions, a *t*-test with two samples and unequal variances was applied. If *p* is less than 0.05 the null hypothesis is rejected, so it is possible to conclude that there is a significant difference between the two samples under analysis.

## 3. Results

### 3.1. Hand-Arm Vibration (HAV)

The effect of vibration on health is duration-dependent and its assessment should be made independently along each axis [38]. Time intervals of 0.4 s were selected in order to obtain the full impact of the ball with the racket and also part of the follow-up phase. The mean and the standard deviation (St.Desv.) of all the values evaluated are presented in Table 4, Table 5, Table 6 and Table 7 and Figure 5.

The HAV values seem to indicate that tennis athletes, during the service, are exposed to levels of effective acceleration that are higher than the action value recommended by the European Directive 2002/44/EC. Actually, Figure 6 compares the evaluated *A*(8) parameter in all axes and the limit values (DEL) and action (DEA) exposures. In this figure, continuous and dashed lines are used to define the DEL and DEA according to the Directive 2002/44/EC, respectively. The total magnitude of vibration (TMV) is also presented in Figure 6.

Figure 6 shows that the values of the TMV parameter exceed the daily exposure limit value in all the situations. No significant differences are observed between the four types of over-grips studied, however, the lowest values are registered for WGWAV and TGWAV. The exposure parameter *A*(8) is a little above the limit value for daily exposure, in the x direction. The number of years of exposure needed until the risk of white finger disease reaches a probability of 10% can also be estimated by Equation (4). The results are presented in Figure 7.

In Figure 7, some differences are observed concerning the diverse types of grips. The number of years of exposure needed until the risk of white finger has a value of around 5 years and 6 months in the case of WGNAV, 6 years and 8 months in the case of CGWAV, and 6 years and 11 months for the other two grips.

### 3.2. Infrared Thermography (IRT)

Table 8 presents the values obtained by Equation (3) for the all athletes concerning the 20 ROIs selected. Regarding Table 8, it is possible to observe that athlete number 6 presents the highest values of *T_sk_* for almost all the ROIs, while athlete number 2 the lowest ones (see also Figure 8).

The contralateral skin temperature of ROIs after the 40 services is obtained according to Equation (6):(6)$$\Delta T_{skRL}=T_{sk,40R}-T_{sk,40L},$$
where $T_{sk,40R}$ is the skin temperature of the right side and $T_{sk,40L}$ is the skin temperature of the left side, respectively, after 40 services. Table 9 shows the results for $\Delta T_{skRL}$, where it is possible to observe that for some athletes the differences recorded are higher than 0.5 °C, which according to Marins et al. [27] is the limit value for not requiring intervention. Table 9 also presents the value of *p*, from statistical analysis, obtained by the *t*-student test.

Figure 9 shows the infrared thermal images of athletes 3 and 5, according to the number of services, while Figure 10 show the variation of skin temperature in all athletes for both RB and LS regions.

Notice that athlete number 1 always presents a negative $\Delta T_{skRL}$ which means that the left side shows a higher final temperature than the right one, which is justified by the fact that this athlete is left-handed (see Figure 11). Athletes 5 and 6 are those who show a lower difference in skin temperature for contralateral ROIs. For athlete 6 the contralateral ROI that indicates the need for monitoring is the extensor carpi radialis brevis, there are other regions with values of 0.5 °C, but most are lower than 0.4 °C. On the other hand, athlete numbers 1 and 2 show a significant quantity of situations that need attention, with some values higher than 1.6 °C. In order to clarify the differences between athletes 1 and 6, Figure 11 shows a comparison of the infrared thermal image between athlete numbers 1 and 6, after 40 services.

It may be seen in Figure 11 that the highest arm skin temperature values are detected near the elbow of the arm that holds the racket, left-arm for the athlete 1 and right-arm for athlete 6. It is also possible to confirm from Figure 12 that athlete number 6, who is more experienced than athlete number 1, does not present such significant differences of the skin temperature as athlete number 1. The differences observed in athlete number 1 between the contralateral muscles may indicate a serious risk for the occurrence of injuries, as mentioned before.

## 4. Discussion

This study set out to examine the effect of two different types of over-grip incorporated into the design of a tennis racket frame design and also the presence of one anti-vibrator device. The study was developed not under highly controlled laboratory conditions, but on the tennis court under quasi-realistic playing conditions. Players were allowed to move freely and without constraints on how forcefully they served. Hence, factors like grip position, grip force, or racket velocity and acceleration at ball impact were not fully controlled. Another potential source of variability may arise from removing and then reattaching the accelerometers to the several rackets, although we carefully monitored the accelerometer placement, we cannot guarantee that positions among tests were the same. Despite these limitations, the results provided useful information on the impact and potential benefits of different types of over-grip on the mechanical vibration behavior of rackets when used by tennis players. The infrared thermography was performed using a thermographic camera that offers high scanning speeds, high imaging resolution, and poses no health problems. The first limitation of the IRT technique is the finite depth of penetration, usually confined to 2–3 mm [39]. Another limitation of this technique is the accuracy of detection of temperature variations on spatial resolution, as well as the reliability varies depending on the analyzed areas [40]. Nevertheless, this preliminary study, performed on a limited number of subjects, provided the basis upon which a subsequent investigation of skin temperature response compared with the electrical activity of muscles, could attempt to infer an indirect estimation of the physical efficiency and/or the training level of athletes.

From the results present in Table 4 and Table 5 it is possible to conclude that the presence of the anti-vibrator reduces the HAV by around 2% to 7% [41]. The athlete who presents the highest values for the total magnitude of the vibration is athlete 6, who has more experience and is better positioned in terms of ranking. These results can be justified by the fact that the force imposed during the tennis service by this athlete could be greater than that imposed by the remaining athletes. Higher grip forces exerted by tennis players are associated with higher vibration transferred to the player arm [42]. In fact, athlete 6 is able to produce more power with his legs, due to the movement created by bending his knees, when compared with the others, thus increasing the vibrations induced by the racket. Further, the athlete who presents the lowest value is number 2, the smallest and lightest one, which confirms the last idea. The same tendency is observed in the case of the use of Tourna and Cork in the over-grip. Moreover, it would be expected that, depending on the axis system defined by the position of the accelerometer in the racket, the highest values for effective acceleration would be observed in the direction of ZZ axis. However, in most trials and athletes the highest value occurs in the XX direction, which can be justified by the way each athlete holds the racket, as well as by the movement he makes with his wrist during the service [42,43].

From Table 4, Table 5, Table 6 and Table 7 it is also possible to conclude that the highest values of HAV are evaluated in the case of the standard Wilson Pro racket (WGNAV) and the lowest for Wilson Pro racket with the anti-vibrator device (WGWAV). Moreover, comparing the variation of HAV values among athletes for each over-grip type, i.e., for each one of the WGNAV and WGWAV conditions, it is possible to see that highest difference appears between athletes 2 and 6, and is about 40% on both over-grip types. The cork over-grip should, theoretically, reduce vibrations due to the material characteristics of cork. However, the HAV values increase by around 6%, which can be explained by the difficulty the athletes had in holding the racket firmly, owing to the greater thickness of this over-grip, which means that the athlete must use greater force to service, which increases the vibrations induced. The Tourna grip, when compared with the standard Wilson Pro, does not present significant differences. According to Marchetti et al. [44] the access point of vibration is the palm of the hand, so the way the athlete handles the racket has a significant influence on the HAV. The average values for the weighted acceleration specified (a_hv_), considering the results of the six athletes and obtained in the four situations evaluated are 42.99 ± 8.4, 41.42 ± 8.2, 41.21 ± 8.5, and 43.01 ± 8.4 m/s^2^, for WGNAV, WGWAV, TGWAV, and CGWAV, respectively. The higher values for the standard deviation are justified by the differences observed between the athletes.

Statistical analysis of the results presented through Table 4, Table 5, Table 6 and Table 7 have shown that the presence of the anti-vibrator and Tourna or Cork over-grips can affect athletes differently: for athlete 1, the presence of cork, when compared to Wilson Pro racket with anti-vibrator, affects the values of RMS in the X and Y directions, with *p* values of 0.043 and 0.0296, respectively; for athlete 2, the RMS Y values of the pair Wilson pro racket with and without anti-vibrator show statistically significant differences, with *p* equal to 0.017. The presence of cork on the grip has also statistical significance differences in all the values of RMS, when compared to WGWAV; in the case of athletes 3 and 4 no statistically significance differences are obtained (*p* > 0.05); for the athletes 5 and 6, the TGWAV changes the HAV values significantly, when compared to the WGWAV, for all the RMS values, with *p* values well below 0.05.

The values of the daily exposure action and the daily exposure limit, presented in Figure 7, were obtained considering a daily duration of 300 s. It is possible to observe that the HAV values obtained during the service are much bigger when compared with those evaluated by the authors in other studies [45,46], which means that the risk of the tennis athletes developing various disorders, like the carpal tunnel syndrome, tingling numbness in the fingers (blood vessels and nerves affected), fingers changing color (blood vessels affected), and loss of manual dexterity (nerves and muscles affected), is higher [47,48]. Moreover, the results of Figure 7 indicate that during the service the tennis athlete may be exposed to health and safety risks, which could be minimized by taking some precautions, for example, using a correct tennis serve technique and an appropriate racket with an adjustable grip. A similar idea was also reported by Ferrara and Cohen [3]. Therefore, according to these results, the risk is very high when tennis athletes play the serve.

The infrared thermography technology can be used to identify the regions that are more solicited during physical activity and exercise and is one of the most often used indicators of health status in humans [49]. From the results presented in Table 8, it may be seen that for almost all the athletes and for all the muscles under analysis, there is a decrease in the skin temperature, which agrees with previous works [25,49,50], and can be explained by the cutaneous vasoconstrictor response to exercise. In fact, the results of Table 8 show that for some athletes the differences are higher than 0.5 °C and it may require attention. Moreover, from the results of the contralateral skin temperature for the different regions of interest presented in Table 9, is possible to see that major variances occur in the RECRB/LECRB contralateral ROIs, followed by the RB/LB contralateral regions, with values that are higher than 0.4 °C in four athletes. These observations may justify the frequent appearance of the injury known as tennis elbow. In fact, Hildebrandt et al. [51] says that a difference of more than 1 °C between contralateral ROIs can indicate a pathophysiological process and Marins et al. [32] concludes that a difference higher than 1.6 °C for the contralateral ROIs can be considered “High Severity”, in the level of attention, and a difference less than 0.5 °C no special attention is required. Regarding the *p* values presented in Table 9, it can be concluded that *p* is smaller than 0.05 for the comparison between RECRB/LECRB, for all the athletes, and for three athletes in the case of RB/LB. Therefore, this means that there is a statistically significant difference between the two contralateral ROIs under analysis, especially the extensor carpi radialis brevis, which confirms the possible appearance of the tennis elbow injury.

In Figure 9, it may be observed that with the increase in the number of services the skin temperatures become more uniform and distinguishable. There is a clear distinction between the first thermographic photograph (reference situation) and the later ones (after performing services), the latter are very similar, suggesting a uniform distribution of temperature. For athlete number 3 an increase in the skin temperature values may be seen, while for athlete number 5 a decrease in the skin temperature is detected. In fact, Figure 10 shows that athlete 3 is the unique athlete who presents an increase in the skin temperature values with the number of services. Nevertheless, the skin temperature values evaluated at the reference condition are in accordance with reported by Webb [52], despite being lower than the traditional 36 °C, when the room temperature is about 15–18 °C. The differences observed in the reference skin temperature between several athletes can be explained by the anthropometric data. In fact, factors relating to individual and personal characteristics can influence the skin temperature, for example, the body mass index [32,53], and athlete number 3 has the lowest body mass index, which may help to justify the results for this athlete. The infrared thermal images presented in Figure 11 allows to confirm that independently of the preferred hand used by the athlete to hold rackets, i.e., right-handed or left-handed, the highest arm skin temperature values are detected near the elbow of the arm that holds the racket.

## 5. Conclusions

In terms of the hand-arm vibration induced, it is possible to conclude that the best situation, to avoid injuries to athlete’s arm, with less vibration transmitted, is the Wilson Pro racket with anti-vibrator (WGWAV), and the worst case is the WGNAV. Therefore, it can be confirmed that the presence of the anti-vibrator influences the vibration values greatly. The presence of Cork and/or Tourna on the racket grip does not have any significant effect on the HAV. The values obtained for the HAV are very high and cannot contribute to diminish the risk of injuries for the tennis athlete. In the case of HAV exposure, the daily level of the total magnitude of vibration exceeded the limit of the daily exposure action value, in all situations, which indicates the need for a rapid intervention. The IRT shows significant differences between the average skin temperatures for contralateral ROIs, which means that this technique can be used to identify the risk of injuries in tennis players.

## Figures and Tables

**Figure 1 ijerph-16-05117-f001:**
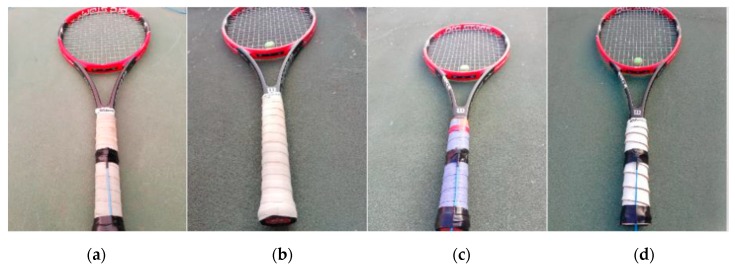
Racket grip test conditions: (**a**) Wilson Pro with without anti-vibrator (WGNAV); (**b**) Wilson Pro with an anti-vibrator (WGWAV); (**c**) Wilson Pro with Tourna and anti-vibrator (TGWAV); (**d**) Wilson Pro with Corck and anti-vibrator CGWAV.

**Figure 2 ijerph-16-05117-f002:**
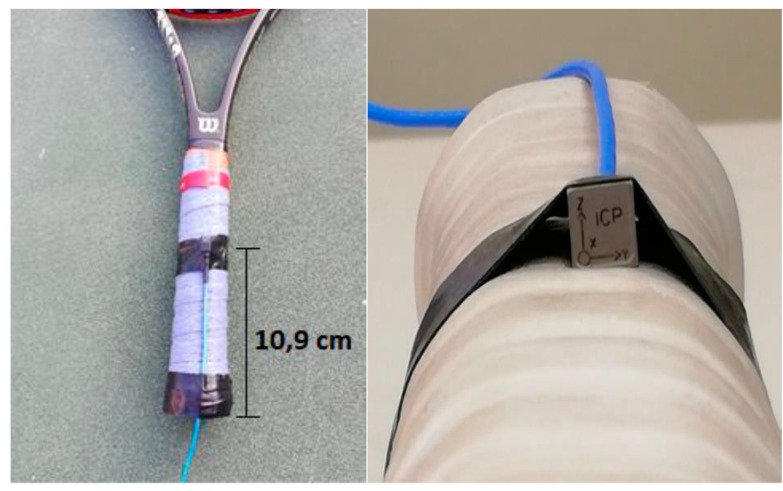
Accelerometer placed in the racket.

**Figure 3 ijerph-16-05117-f003:**
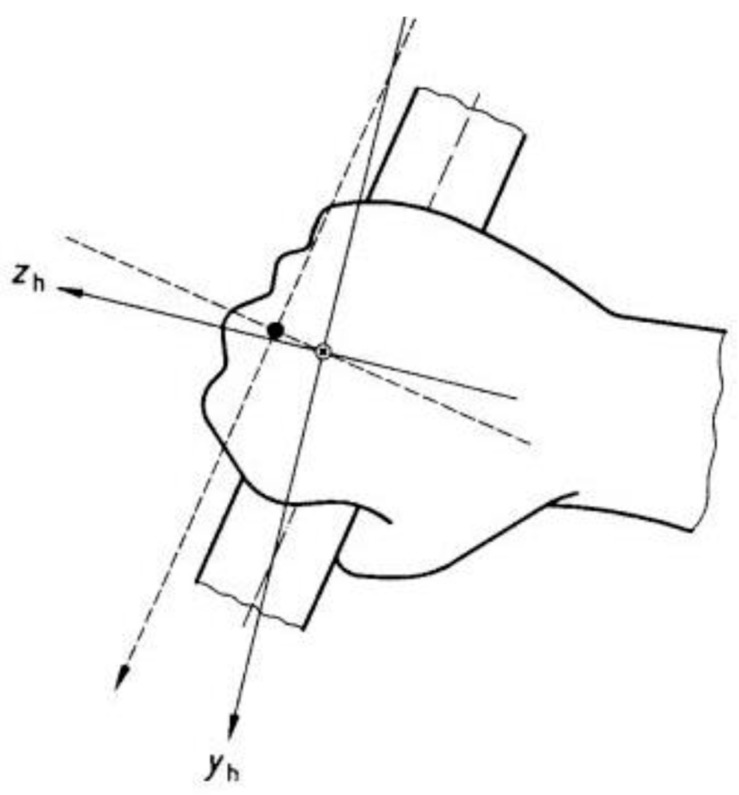
Coordinate systems used in the hand-arm vibration analysis [31].

**Figure 4 ijerph-16-05117-f004:**
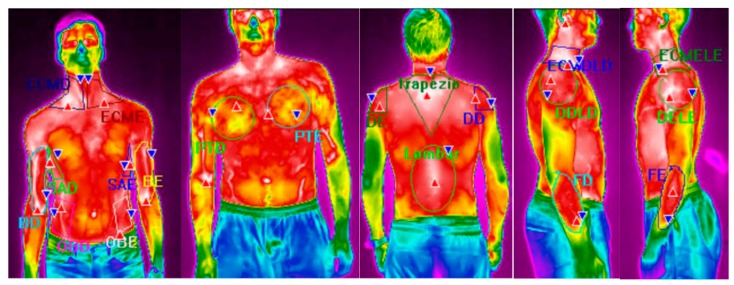
Infrared thermal images of the regions of interest (ROIs).

**Figure 5 ijerph-16-05117-f005:**
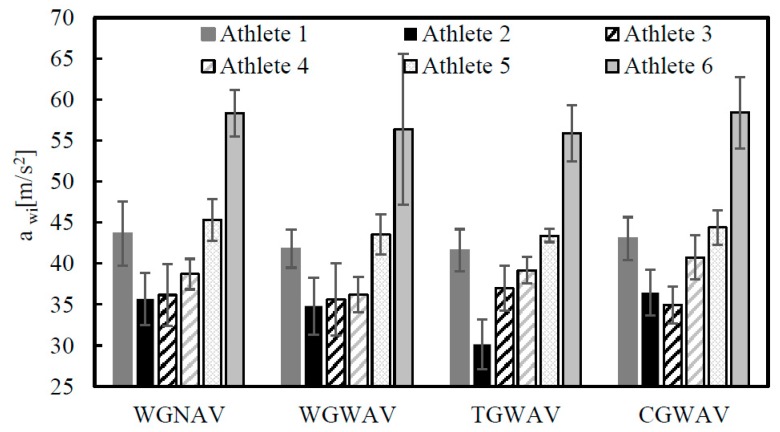
Root mean square acceleration for all athletes and cases studied.

**Figure 6 ijerph-16-05117-f006:**
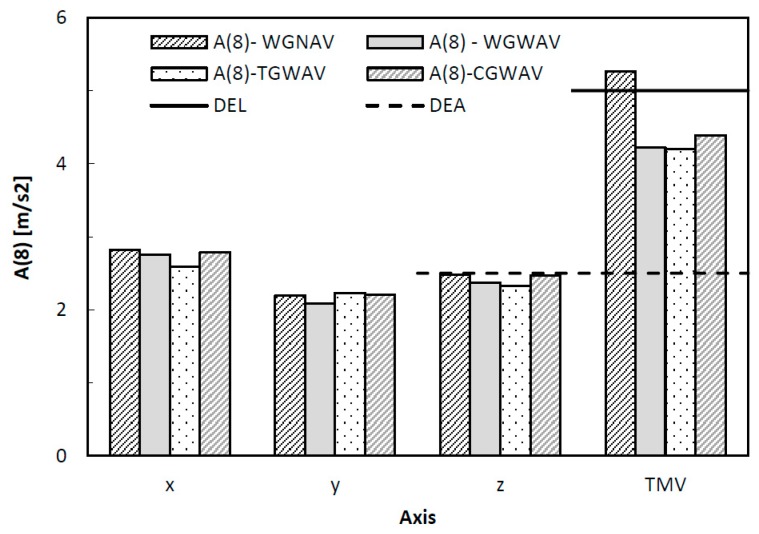
Equivalent 8-h workday acceleration value, *A*(8).

**Figure 7 ijerph-16-05117-f007:**
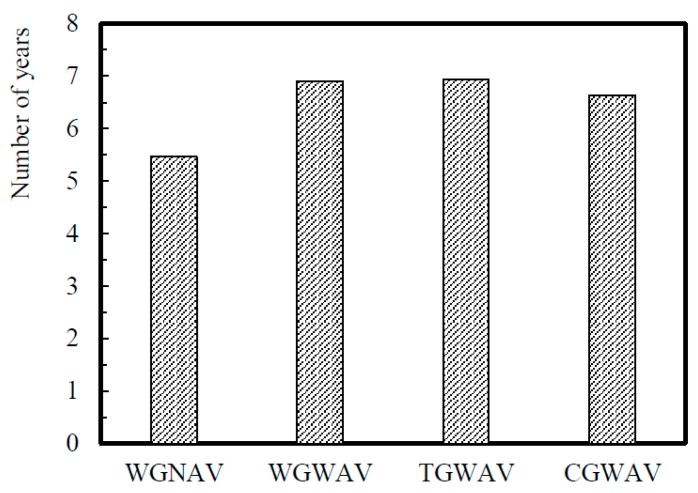
Number of years needed for a 10% possibility of white finger disease: parameter *D*.

**Figure 8 ijerph-16-05117-f008:**
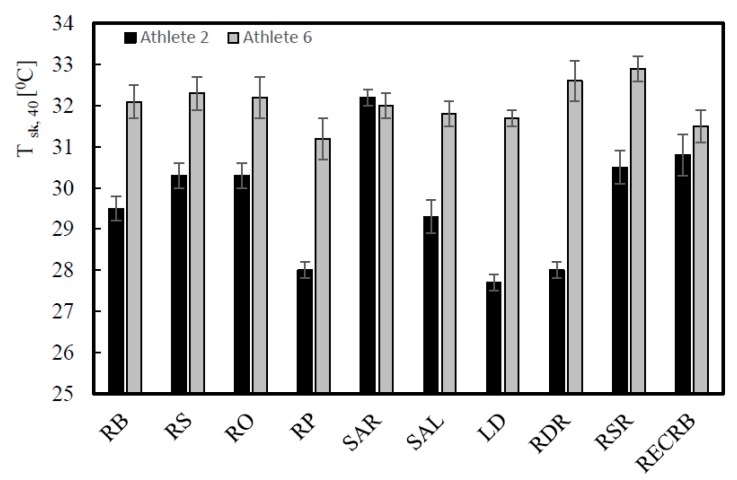
Skin temperature after 40 services at some regions of interest to athletes 2 and 6.

**Figure 9 ijerph-16-05117-f009:**
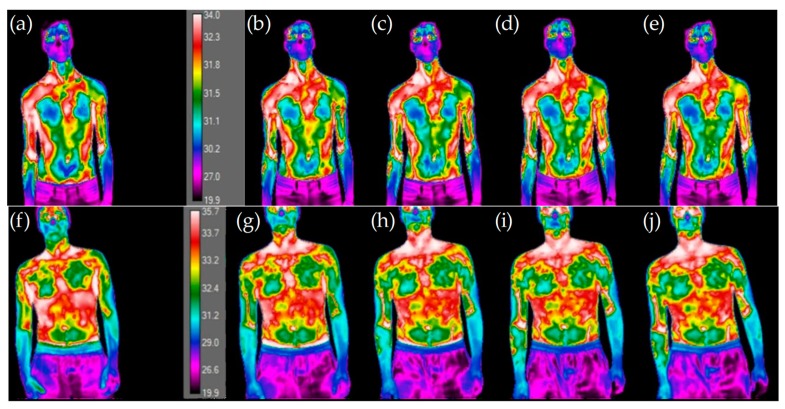
Frontal perspective of the infrared thermal images for athlete number 3: (**a**) reference; (**b**) after 10 services; (**c**) after 20 services; (**d**) after 30 services; (**e**) after 40 services. For athlete number 5: (**f**) reference; (**g**) after 10 services; (**h**) after 20 services; (**i**) after 30 services; (**j**) after 40 services.

**Figure 10 ijerph-16-05117-f010:**
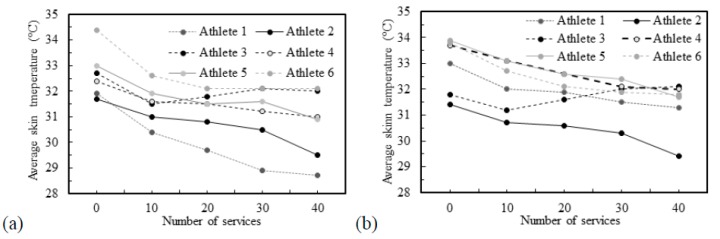
Average skin temperature in all athletes: (**a**) right biceps (RB); (**b**) left sternocleidomastoid (LS).

**Figure 11 ijerph-16-05117-f011:**
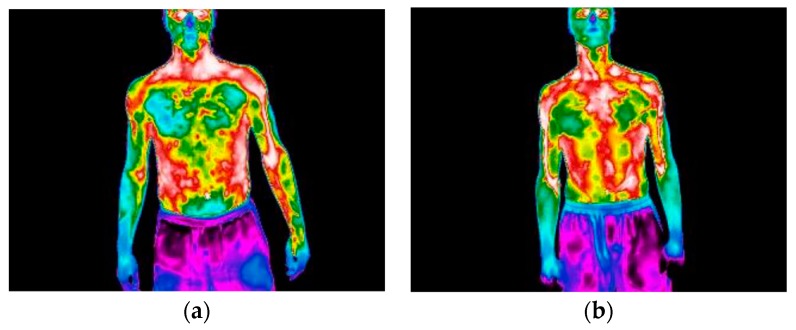
Infrared thermal images after 40 services: (**a**) athlete number 1; (**b**) athlete number 6.

**Figure 12 ijerph-16-05117-f012:**
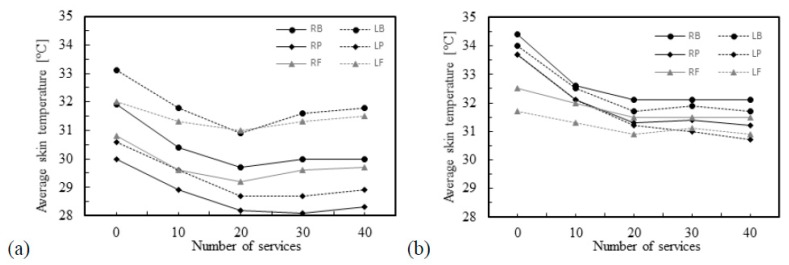
Average skin temperature of several regions of interest in the athletes: (**a**) number 1; (**b**) number 6.

**Table 1 ijerph-16-05117-t001:** Volunteers’ relevant characteristics.

Athlete’s Number	Age	Years of Practice	Height (cm)	Weight (kg)	Wingspan (cm)	Body Mass	Preferred Hand
1	19	10	174	72	174	23.7	Left
2	21	10	173	65	172	21.7	Right
3	21	13	185	70	185	20.4	Right
4	23	13	177	66	171	21.1	Right
5	21	14	182	84	184	25.3	Right
6	22	14	183	72	183	21.4	Right

**Table 2 ijerph-16-05117-t002:** Randomized order.

Athlete Number	1st Test	2nd Test	3rd Test	4th Test
1	WGNAV	WGWAV	TGWAV	CGWAV
2	CGWAV	WGNAV	WGWAV	TGWAV
3	TGWAV	CGWAV	WGNAV	WGWAV
4	WGWAV	WGNAV	CGWAV	TGWAV
5	TGWAV	WGWAV	WGNAV	CGWAV
6	CGWAV	TGWAV	WGWAV	WGNAV

**Table 3 ijerph-16-05117-t003:** Protocol for the infrared thermography (IRT) evaluation.

Experimental Methodology for IRT	
Time	Action
8 min	Sit completely immobile in a chair
1 min	Thermographic photos
1 min	Executed 10 services
1 min	Thermographic photos
1 min	Executed 10 services
1 min	Thermographic photos
1 min	Executed 10 services
1 min	Thermographic photos
1 min	Executed 10 services
1 min	Thermographic photos

**Table 4 ijerph-16-05117-t004:** Directional root means square accelerations in the case of Wilson Pro racket without an anti-vibrator (WGNAV).

Athlete	x (m/s^2^)		y (m/s^2^)		z (m/s^2^)		$a_{w}\;(m/s^{2})$	St.Desv (m/s^2^)
	RMS	St.Desv.	RMS	St.Desv.	RMS	St.Desv.		
1	29.51	3.75	16.43	1.79	27.69	2.31	43.68	3.90
2	20.93	2.67	18.68	1.87	22.08	2.39	35.70	3.17
3	24.96	2.95	16.92	3.33	19.98	2.33	36.17	3.75
4	25.78	3.24	19.84	1.48	21.01	2.86	38.72	1.86
5	34.34	3.14	20.34	1.62	21.48	2.18	45.32	2.53
6	30.23	2.85	36.74	3.89	33.74	4.44	58.33	2.83

**Table 5 ijerph-16-05117-t005:** Directional root means square accelerations in the case of Wilson Pro racket with an anti-vibrator (WGWAV).

Athlete	x (m/s^2^)		y (m/s^2^)		z (m/s^2^)		$a_{w}\;(m/s^{2})$	St.Desv. (m/s^2^)
	RMS	St.Desv.	RMS	St.Desv.	RMS	St.Desv.		
1	27.01	1.95	17.19	2.53	26.93	3.17	41.83	2.33
2	23.55	2.55	15.65	2.27	20.36	2.24	34.84	3.47
3	22.59	4.25	17.53	2.81	21.33	1.71	35.67	4.40
4	23.91	2.25	18.71	0.94	19.74	1.23	36.21	2.19
5	34.47	1.56	18.62	2.46	19.06	1.31	43.57	2.48
6	30.69	4.70	34.82	1.51	32.06	1.39	56.41	9.21

**Table 6 ijerph-16-05117-t006:** Directional root means square accelerations in the case of Wilson Pro racket with an anti-vibrator and the Tourna over-grip (TGWAV).

Athlete	x (m/s^2^)		y (m/s^2^)		z (m/s^2^)		$a_{w}\;(m/s^{2})$	St.Desv. (m/s^2^)
	RMS	St.Desv.	RMS	St.Desv.	RMS	St.Desv.		
1	25.62	2.32	17.09	2.84	27.99	2.60	41.62	2.58
2	19.53	1.19	14.13	1.67	18.12	1.42	30.16	3.06
3	26.84	3.63	17.35	2.25	18.62	0.76	36.99	2.76
4	26.95	2.29	19.95	2.91	20.29	1.29	39.19	1.63
5	31.61	1.97	23.17	1.05	18.69	1.54	43.42	0.79
6	21.73	2.49	39.53	2.69	33.01	2.07	55.89	3.44

**Table 7 ijerph-16-05117-t007:** Directional root means square accelerations in the case of Wilson Pro racket with an anti-vibrator and the Cork over-grip (CGWAV).

Athlete	x (m/s^2^)		y (m/s^2^)		z (m/s^2^)		$a_{w}\;(m/s^{2})$	St.Desv. (m/s^2^)
	RMS	St.Desv.	RMS	St.Desv.	RMS	St.Desv.		
1	29.75	1.01	14.66	1.64	27.44	2.57	43.05	2.62
2	21.07	1.97	19.25	2.66	22.68	0.99	36.45	2.76
3	22.91	2.66	17.11	2.48	20.07	1.69	34.94	2.22
4	27.72	2.17	21.01	1.81	21.27	1.57	40.77	2.69
5	35.21	3.34	20.35	2.04	17.79	1.52	44.39	2.10
6	27.23	4.19	37.33	1.92	35.73	3.49	58.41	4.38

**Table 8 ijerph-16-05117-t008:** Thermal skin variations for the different regions of interest.

ROIS		Athlete 1	Athlete 2	Athlete 3	Athlete 4	Athlete 5	Athlete 6
RB	$T_{sk,40}$	30.0 ± 0.3	29.5 ± 0.3	32.3 ± 0.3	31.0 ± 0.4	30.9 ± 0.4	32.1 ± 0.4
	$\Delta T_{sk}$	−1.9	−2.2	−0.7	−1.4	−2.1	−2.3
LB	$T_{sk,40}$	31.8 ± 0.5	27.9 ± 0.2	32.0 ± 0.30	30.4 ± 0.2	30.2 ± 0.3	31.7 ± 04
	$\Delta T_{sk}$	−1.3	−2.8	2	−2.0	−2.0	−2.3
RS	$T_{sk,40}$	31.3 ± 0.2	30.3 ± 0.3	33.1 ± 0.30	33.2 ± 0.4	31.9 ± 0.4	32.3 ± 0.4
	$\Delta T_{sk}$	−1.7	−1.4	6	−1.1	−2.1	−1.7
LS	$T_{sk,40}$	31.9 ± 0.5	29.4 ± 0.2	32.1 ± 0.30	32.0 ± 0.3	31.7 ± 0.4	31.8 ± 0.3
	$\Delta T_{sk}$	−1.1	−2.0	3	−1.7	−2.2	−2.0
RO	$T_{sk,40}$	30.9 ± 0.3	30.3 ± 0.3	31.8 ± 0.4	32.3 ± 0.6	30.5 ± 0.4	32.2 ± 0.5
	$\Delta T_{sk}$	−1.4	−1.5	−0.7	−2.0	−3.5	−2.7
LO	$T_{sk,40}$	31.0 ± 0.4	29.4 ± 0.3	31.0 ± 0.30	32.0 ± 0.4	30.3 ± 0.4	31.9 ± 0.5
	$\Delta T_{sk}$	−1.7	−1.5	0	−1.8	−3.3	−1.9
RP	$T_{sk,40}$	28.3 ± 0.3	28.0 ± 0.2	31.5 ± 0.4	30.5 ± 0.4	30.9 ± 0.3	31.2 ± 0.5
	$\Delta T_{sk}$	−1.7	−0.2	−0.4	−3.0	−3.5	−3.0
LP	$T_{sk,40}$	28.9 ± 0.2	27.4 ± 0.1	30.9 ± 0.4	29.8 ± 0.2	30.1 ± 0.4	30.7 ± 0.3
	$\Delta T_{sk}$	−1.7	−0.5	−0.3	−2.7	−3.0	−2.5
SAR	$T_{sk,40}$	30.7 ± 0.3	32.2 ± 0.2	32.2 ± 0.3	32.1 ± 0.4	30.7 ± 0.4	32.0 ± 0.3
	$\Delta T_{sk}$	−1.9	−0.6	−0.7	−2.1	−3.8	−3.0
SAL	$T_{sk,40}$	31.1 ± 0.4	29.3 ± 0.4	30.8 ± 0.3	32.0 ± 0.2	30.6 ± 0.2	31.8 ± 0.3
	$\Delta T_{sk}$	−1.7	−0.8	−0.6	−1.7	−3.7	−2.2
RD	$T_{sk,40}$	28.9 ± 0.2	27.9 ± 0.2	31.6 ± 0.4	30.7 ± 0.3	30.3 ± 0.4	31.2 ± 0.3
	$\Delta T_{sk}$	−1.9	−0.4	−0.3	−2.5	−2.8	−2.5
LD	$T_{sk,40}$	29.8 ± 0.3	27.7 ± 0.2	32.0 ± 0.3	30.9 ± 0.4	30.6 ± 0.3	31.7 ± 0.2
	$\Delta T_{sk}$	−1.7	−0.3	0.1	−2.5	−2.3	−2.3
L	$T_{sk,40}$	29.4 ± 0.3	28.0 ± 0.3	31.5 ± 0.4	30.5 ± 0.4	30.9 ± 0.3	31.7 ± 0.5
	$\Delta T_{sk}$	−2.4	−1.4	−0.4	−3.4	−3.4	−2.8
T	$T_{sk,40}$	31.3 ± 0.4	29.2 ± 0.2	32.3 ± 0.3	32.0 ± 0.3	31.7 ± 0.4	32.7 ± 0.4
	$\Delta T_{sk}$	−1.3	−0.9	−0.2	−2.1	−3.1	−2.3
RDR	$T_{sk,40}$	30.0 ± 0.2	28.0 ± 0.2	31.8 ± 0.4	30.4 ± 0.4	30.0 ± 0.2	32.6 ± 0.5
	$\Delta T_{sk}$	−1.4	0.1	−0.2	−2.8	−2.8	−1.5
LDL	$T_{sk,40}$	30.7 ± 0.4	28.2 ± 0.4	32.1 ± 0.3	30.7 ± 0.5	30.2 ± 0.3	32.2 ± 0.4
	$\Delta T_{sk}$	−1.3	0.70	0.4	−2.5	−2.8	−1.4
RSR	$T_{sk,40}$	30.7 ± 0.2	30.5 ± 0.4	32.9 ± 0.4	33.0 ± 0.4	32.3 ± 0.4	32.9 ± 0.3
	$\Delta T_{sk}$	−1.5	−0.8	0.4	−1.4	−1.1	−0.9
LSL	$T_{sk,40}$	31.8 ± 0.4	30.4 ± 0.2	32.5 ± 0.4	33.0 ± 0.5	32.2 ± 0.2	32.6 ± 0.4
	$\Delta T_{sk}$	−0.6	−1.1	0.5	−1.2	−1.6	−1.6
RECRB	$T_{sk,40}$	29.7 ± 0.4	30.8 ± 0.5	32.5 ± 0.5	31.8 ± 0.4	31.5 ± 0.3	31.5 ± 0.4
	$\Delta T_{sk}$	−1.3	0.7	0.4	−2.5	−2.8	−1.4
LECRB	$T_{sk,40}$	31.5 ± 0.4	29.9 ± 0.4	31.1 ± 0.3	31.2 ± 0.5	29.9 ± 0.3	30.9 ± 0.4
	$\Delta T_{sk}$	−0.5	0.3	0.0	−0.5	−1.8	−0.8

**Table 9 ijerph-16-05117-t009:** Contralateral skin temperature for the different regions of interest.

ROIS	Athlete 1	Athlete 2	Athlete 3	Athlete 4	Athlete 5	Athlete 6
RB/LB	−1.8	1.6	0.3	0.6	0.7	0.4
	*p* < 0.05 **	*p* < 0.05 **	*p* > 0.05	*p* > 0.05	*p* > 0.05	*p* < 0.05 **
RS/LS	−0.6	0.9	1.0	1.2	0.2	0.5
	*p* > 0.05	*p* > 0.05	*p* < 0.05 **	*p* > 0.05	*p* > 0.05	*p* > 0.05
RO/LO	−0.1	0.9	0.8	0.3	0.2	0.3
	*p* > 0.05	*p* > 0.05	*p* > 0.05	*p* > 0.05	*p* > 0.05	*p* > 0.05
RP/LP	−0.6	0.6	0.6	0.7	0.8	0.5
	*p* > 0.05	*p* > 0.05	*p* = 0.05	*p* > 0.05	*p* > 0.05	*p* > 0.05
SAR/SAL	−0.4	2.9	1.4	0.1	0.1	0.2
	*p* > 0.05	*p* > 0.05	*p* > 0.05	*p* > 0.05	*p* > 0.05	*p* > 0.05
RD/LD	−0.9	0.2	−0.4	−0.2	−0.3	−0.5
	*p* > 0.05	*p* < 0.05	*p* > 0.05	*p* > 0.05	*p* > 0.05	*p* > 0.05
RDR/LDL	−0.7	−0.2	−0.3	−0.3	−0.2	0.4
	*p* > 0.05	*p* > 0.05	*p* > 0.05	*p* > 0.05	*p* > 0.05	*p* > 0.05
RSR/LSL	−1.1	0.1	0.4	0	0.1	0.3
	*p* > 0.05	*p* > 0.05	*p* > 0.05	*p* > 0.05	*p* > 0.05	*p* > 0.05
RECRB/LECRB	−1.8	0.9	1.4	0.6	1.6	0.6
	*p* < 0.05 **	*p* < 0.05 **	*p* < 0.05 **	*p* < 0.05 **	*p* < 0.05 **	*p* < 0.05 **

** there is a statistically significant difference.

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
