# Peer review of "Hand-Arm Vibration Assessment and Changes in the Thermal Map of the Skin in Tennis Athletes during the Service"

_ijerph, 2019, doi:10.3390/ijerph16245117_

Round 1
Reviewer 1 Report
Dear Author,
As a Whole the study is thorough and well-written and the theme is of course of interest for both Professional and amateur tennis players, since epicondylitis is a well-known secondary complication.
It is a small case study and the results can possibly offer some advice to the players and to those who develop rackets, but generalisations should not be made on such small samples of the population.
I find the study a bit "Heavy" to read (Language) and lengthy so the text could be a bit Shorter and more stringent in its form. The figures and tables are not self-explanatory and it is preferable not to use abreviations in the title text.Table 8 could in its present form be excluded. Please Review the amount of figures as well. Figures and tables should ADD to the understanding of the text so you need to describe and choose wisely what to use.
It is not easy to follow the Logic in the Results / discussion section and I would have preferred a division here. Results and Discussion separately. Perhaps check instructions for Authors.
The infra red light as an indicator (predictor?) of inflammation of muscles should also be discussed. And the weaknesses and strengths With the study.
This should be reflected in the Conclusion, which could preferably should be shortened, summing up the main results, not presenting any.
Author Response
Response to the reviewer’s comments:
The authors would like to thank the reviewers for reading the manuscript and for the comments and suggestions to improve the manuscript. Changes in the revised manuscript are highlighted with blue color.
Reviewer 1
Paper: Hand-arm vibrations assessment and changes in the thermal map of the skin in tennis athletes during the service
As a Whole the study is thorough and well-written and the theme is of course of interest for both Professional and amateur tennis players, since epicondylitis is a well-known secondary complication.
Authors response: We agree with the reviewer.
It is a small case study and the results can possibly offer some advice to the players and to those who develop rackets, but generalizations should not be made on such small samples of the population.
Authors response: We agree with the reviewer and, therefore, we added the following statement at the end of discussion section: “Nevertheless, this preliminary study, performed on a limited number of subjects, provided the basis upon which a subsequent investigation of skin temperature response compared with the electrical activity of muscles, could attempt to infer an indirect estimation of the physical efficiency and/or the training level of athletes.”
I find the study a bit "Heavy" to read (Language) and lengthy so the text could be a bit Shorter and more stringent in its form.
Authors response: we did some changes in the arrangement of text, in order to clarify the main points.
The figures and tables are not self-explanatory and it is preferable not to use abbreviations in the title text.
Authors response: All figure and tables captions were changed in order to remove abbreviations.
Table 8 could in its present form be excluded. Please Review the amount of figures as well. Figures and tables should ADD to the understanding of the text so you need to describe and choose wisely what to use.
Authors response: Table 8 was removed from the paper, but the text related with this table was maintained. We also have reduced the number of figures to 13. In our opinion, all these figures are important to help readers to understand the work.
It is not easy to follow the Logic in the Results / discussion section and I would have preferred a division here. Results and Discussion separately. Perhaps check instructions for Authors.
Authors response: we agree with reviewer. The paper was changed in order to address this suggestion and we separated Results and Discussion sections.
The infrared light as an indicator (predictor?) of inflammation of muscles should also be discussed. And the weaknesses and strengths With the study.
Authors response: Actually, there are some published works wherein they check the viability of this technique to detect skin pathologies, but muscles infections is not usual. Therefore, at the end of discussion section the following additional text was added: “The infrared thermography was performed using a thermographic camera that offers high scanning speeds, high imaging resolution and pose no health problems. The first limitation of the IRT technique is the finite depth of penetration, usually confined to 2-3 mm [49]. Another limitation of this technique is the accuracy of detection of temperature variations on spatial resolution, as well as the reliability varies depending on the analyzed areas [50]. Nevertheless, this preliminary study, performed on a limited number of subjects, provided the basis upon which a subsequent investigation of skin temperature response compared with the electrical activity of muscles, could attempt to infer an indirect estimation of the physical efficiency and/or the training level of athletes.”
This should be reflected in the Conclusion, which could preferably should be shortened, summing up the main results, not presenting any.
Authors response: the conclusion was shortened, by removing some explanatory text.

Reviewer 2 Report
Dear Editor, dear Authors,
Thank you for giving me the opportunity to evaluate the manuscript: “Hand-arm vibrations assessment and changes in the thermal map of the skin in tennis athletes during the service”, ijerph-647229.
This work presents a biomechanical analysis of tennis players with reference to injuries reported at the level of the elbow. The analysis is based on vibration analysis and thermography.
Overall the paper seems well written. The paper is interesting as the quantitative analysis of performance/sport actions is currently a topic of extensive research.
I believe that there are some methodological issues that needs to be fixed before the paper can be accepted. More details in the following.
The introduction proposes a literature review mainly focused on sport injuries and their causes. I would recommend the authors to review and discuss also the technical aspects i.e. the methods commonly used for motion capture/analysis, vibration analysis, measurement of human performance, etc. Some suggestions are in the following papers:
Stereophotogrammetry in Functional Evaluation: History and Modern Protocols. In SpringerBriefs in Applied Sciences and Technology; 2018; pp. 1–29. DOI: 10.1007/978-3-319-67437-7_1
Stride and Step Length Obtained with Inertial Measurement Units during Maximal Sprint Acceleration, Sports 2019, 7(9), 202; DOI: 10.3390/sports7090202
Indirect Measurement of Ground Reaction Forces and Moments by Means of Wearable Inertial Sensors: A Systematic Review. Sensors 2018, 18, 2564. DOI: 10.3390/s18082564
Hand-held Dynamometry Correlation With the Gold Standard Isokinetic Dynamometry: A Systematic Review, PM&R, DOI: 10.1016/j.pmrj.2010.10.025.
Analysis of Knee Strength Measurements Performed by a Hand-Held Multicomponent Dynamometer and Optoelectronic System. IEEE Trans. Instrum. Meas. 2017, 66, 85–92. DOI: 10.1109/TIM.2016.2620799.
Stride length determination during overground running using a single foot-mounted inertial measurement unit. Journal of biomechanics 2018, DOI: 10.1016/j.jbiomech.2018.02.003.
The authors used a wired accelerometer. In general, wireless sensors are to be preferred to reduce noise/artifacts due to the cable. (Of course there are pros of wired sensors). I would recommend the author to better discuss this design choice.
Please describe if some noise filtering or other pre processing were applied to raw data.
The authors used thermography to detect signs of injury. Please improve the description (both in the introduction and discussion) of the rationale behind the clinical use of thermography. (What was written at line 337 is not enough). In other words: describe the clinical validation/applicability of thermography for such measurements, how the superficial temperature correlates to fatigue/injury and the reliability of this technique to this scope.
The authors cite a European Normative on vibration that applies mainly to workers. It would be interesting to deepen to what extent this normative can be applied to sports, i.e. how the DEA and DEL applies to sports. I would recommend the authors to elaborate this part.
In the table/figure captions please explain the acronyms used.
Figure 5: I don’t understand the meaning of “amplitude on [Hz]”.
Line 242: It is not clear how those values were deducted from the tables.
Please mark the statistically significant differences with a * in graphs and tables.
Regarding the analysis of acceleration, instead of considering x,y,z, axes independently, I would recommend to use the standard vector magnitude.
It would be interesting to have a spectral analysis of the recorded vibration and/or the peak vibration frequency for the athletes.
Please improve the discussion about how the vibration correlates to the elbow injuries and briefly discuss the limitations of this study and possible future works.
Author Response
Reviewer 2
Overall the paper seems well written. The paper is interesting as the quantitative analysis of performance/sport actions is currently a topic of extensive research
Authors response: we agree, Thanks!
The introduction proposes a literature review mainly focused on sport injuries and their causes. I would recommend the authors to review and discuss also the technical aspects i.e. the methods commonly used for motion capture/analysis, vibration analysis, measurement of human performance, etc. Some suggestions are in the following papers:
Stereo photogrammetry in Functional Evaluation: History and Modern Protocols. In Springer Briefs in Applied Sciences and Technology; 2018; pp. 1–29. DOI: 10.1007/978-3-319-67437-7_1
Stride and Step Length Obtained with Inertial Measurement Units during Maximal Sprint Acceleration, Sports 2019, 7(9), 202; DOI: 10.3390/sports7090202
Indirect Measurement of Ground Reaction Forces and Moments by Means of Wearable Inertial Sensors: A Systematic Review. Sensors 2018, 18, 2564. DOI: 10.3390/s18082564
Hand-held Dynamometry Correlation With the Gold Standard Isokinetic Dynamometry: A Systematic Review, PM&R, DOI: 10.1016/j.pmrj.2010.10.025.
Analysis of Knee Strength Measurements Performed by a Hand-Held Multicomponent Dynamometer and Optoelectronic System. IEEE Trans. Instrum. Meas. 2017, 66, 85–92. DOI: 10.1109/TIM.2016.2620799.
Stride length determination during over-ground running using a single foot-mounted inertial measurement unit. Journal of biomechanics 2018, DOI: 10.1016/j.jbiomech.2018.02.003.
Authors response: we added some of these references within introduction, it now can be read as: “Nowadays, there are a relatively great number of techniques and studies that include motion capture/analysis, vibration analysis, measurement of human performance, etc. [20–23] but there are few studies using infrared thermography (IRT).”
The authors used a wired accelerometer. In general, wireless sensors are to be preferred to reduce noise/artifacts due to the cable. (Of course, there are pros of wired sensors). I would recommend the author to better discuss this design choice.
Authors response: In fact, the wireless accelerometer can be used in this application, however, we are a research team that has some limitations on the budget and, therefore, this accelerometer was the only one available for this work.
Please describe if some noise filtering or other pre-processing were applied to raw data.
Authors response: yes, we added the following text in the material and methods: “The acceleration data was filtered, as specified in the standards, using the SVT Human Vibration tool of LabVIEW (LabView, Manual of NI Sound and Vibration Measurement Suite 6.0, 2007). The filter coefficients were evaluated according to the procedures developed by Rimmel and Mansfield [37].”
The authors used thermography to detect signs of injury. Please improve the description (both in the introduction and discussion) of the rationale behind the clinical use of thermography. (What was written at line 337 is not enough). In other words: describe the clinical validation/applicability of thermography for such measurements, how the superficial temperature correlates to fatigue/injury and the reliability of this technique to this scope.
Authors response: in order to detect the fatigue/injury of muscles it will be preferable to use the electromyography, but the best option it will be to correlated both techniques or even three techniques to try to get some conclusions. Hence, at the end of discussion section we added the following text: “The first limitation of the IRT technique is the finite depth of penetration, usually confined to 2-3 mm [54]. Another limitation of this technique is the accuracy of detection of temperature variations on spatial resolution, as well as the reliability varies depending on the analyzed areas [55]. Nevertheless, this preliminary study, performed on a limited number of subjects, provided the basis upon which a subsequent investigation of skin temperature response compared with the electrical activity of muscles, could attempt to infer an indirect estimation of the physical efficiency and/or the training level of athletes.”
The authors cite a European Normative on vibration that applies mainly to workers. It would be interesting to deepen to what extent this normative can be applied to sports, i.e. how the DEA and DEL applies to sports. I would recommend the authors to elaborate this part.
Authors response: the following text was added to discussion: “Since shock and vibration were recognized as a source of injuries at the workplace, the ergonomic intervention over the years has made significant strides to minimize that issues in many jobs. Nevertheless, the effects of shock and vibration has also become to be considered as a significant risk factor in several professional sports, including baseball, cross motorcycle, soccer, football and tennis [9,10,39]. In fact, this interest results from a wide variety of sport pathologies related with the head, heart and the musculoskeletal system that have been reported in the literature [9,40]. Moreover, some of these pathologies can be responsible for significant disability, impairment of function and careers, financial cost and post-career disease, including even death. Although the major attention has been focused on professional sports, similar problems have also been identified in the amateur arena of the same sports [41]. Hence, in authors opinion, the scientific community should be committed to address and resolve this risk, assuring that a multidisciplinary effort, similar to that used to control whole body and hand-arm vibration in the workplace, can be also instituted at sports. Meanwhile, because the only standard that analyze the vibrations effect in the human is the ISO 5349, the authors have no other option than use this stand to estimate hand-arm vibrations and the European Directive 2002/44/EC to quantify the vibration exposure in tennis players.
Furthermore, there are a lot of studies concerning the injuries in tennis athletes, but, according to these authors’ knowledge, no studies have been done to evaluate the HAV, simultaneously with the IRT, to identify the probability of injury in tennis athletes during the service, until now. Actually, during the review process of this work, it was possible to identify the work of Yeh et al. [42] wherein racket vibration behavior was examined with a novel vibration damping technology (VDT) into tennis forehand and serve techniques. The only difference in the VDT racket was the inclusion of a layer of composite material in the frame. The composite materials combine a traditional damping layer with a layer of fibre preform. Nevertheless, the authors quantify acceleration levels using a methodology that evaluates the peak acceleration, for each stroke, and, subsequently, the mean peak acceleration across ten strokes. The level of acceleration reported by these authors is even higher than that evaluated in our work, but, even though, they confirm that vibration dampening technologies can effectively reduce mechanical racket vibration after ball impact”
In the table/figure captions please explain the acronyms used.
Authors response: all the acronyms were removed and changed according the suggestions.
Figure 5: I don’t understand the meaning of “amplitude on [Hz]”.
Authors response: it now reads: “Figure 5. Data collected from all the three axes with amplitude on [m/s2].”
Line 242: It is not clear how those values were deducted from the tables.
Authors response: it now reads: “Moreover, comparing the variation of HAV values among athletes for each over-grip type, i.e. for each one of WGNAV and WGWAV conditions, it is possible to see that the highest difference appears between athletes 2 and 6, and is about 40% on both over-gripe types.”
Please mark the statistically significant differences with a * in graphs and tables.
Authors response: They are identified at bold, but and additional identification was considered.
Regarding the analysis of acceleration, instead of considering x,y,z, axes independently, I would recommend to use the standard vector magnitude.
Authors response: As explained in section 3.1, the effect of vibration on health is duration-dependent and its assessment should be made independently along each axis [38]. Moreover, the ISO 5349 standards specify that the total magnitude of the vibration should be determined from the vibration in the (x,y,z) orthogonal coordinates and using equation (3) of the paper.
It would be interesting to have a spectral analysis of the recorded vibration and/or the peak vibration frequency for the athletes.
Authors response: in the case of WBV (wall body vibrations) and for certain types of vibrations, especially for those containing occasional shocks, the basic evaluation method may underestimate the effects of vibration. So, in order to verify the suitability of the weighted rms acceleration method to those types of vibration, the ISO Standard uses the crest factor (CF), which is defined as the modulus of the ratio of the maximum instantaneous peak value of the frequency-weighted acceleration signal to its rms value [ see reference 39]. In these cases, the ISO 2631 should be used. Nevertheless, in HAV standard it is not require to perform a frequency spectrum analysis.
Please improve the discussion about how the vibration correlates to the elbow injuries and briefly discuss the limitations of this study and possible future works.
Authors response: The following text was added to describe the limitations: “This study set out to examine the effect of two different types of over-grip incorporated into the design of a tennis racket frame design and also the presence of one anti-vibrator device. The study was developed not under highly controlled laboratory conditions, but on the tennis court under quasi-realistic playing conditions. Players were allowed to move freely and without constraints on how forceful they put on serves. Hence, factors like grip position, grip force, or racket velocity and acceleration at ball impact were not fully controlled. Another potential source of variability may arise from removing and then re-attaching the accelerometers to the several rackets, although we carefully monitored of accelerometer placement, we cannot guarantee that positions among tests were the same. Despite these limitations, the results provided a useful information on the impact and potential benefits of different types of over-grip on the mechanical vibration behavior of racket when used by tennis players. The infrared thermography was performed using a thermographic camera that offers high scanning speeds, high imaging resolution and pose no health problems. The first limitation of the IRT technique is the finite depth of penetration, usually confined to 2-3 mm [54]. Another limitation of this technique is the accuracy of detection of temperature variations on spatial resolution, as well as the reliability varies depending on the analyzed areas [55]. Nevertheless, this preliminary study, performed on a limited number of subjects, provided the basis upon which a subsequent investigation of skin temperature response compared with the electrical activity of muscles, could attempt to infer an indirect estimation of the physical efficiency and/or the training level of athletes.”

Reviewer 3 Report
General Comments to Author:
This paper conducted Hand-arm vibrations (HAV) assessment and monitored the changes in the thermal map of the skin in tennis athletes. In general, the paper is well written and easy-to-follow, with significant application prospect. The reviewer would suggest following few minor changes that could be made to improve the quality of the paper.
Specific Comments to Author:
Literature review could be more focused. Maybe there lacks detailed explanations about the key contributions. The readers need more help to understand what is important, what is new, and how it relates to the state of art. Some figures need improvement. Specifically, to my point of view, Figure 5 is not a good representative of the others. In “Results and discussion” section, authors give some figures. However, they do not make strong conclusions. What are the implications of the findings? More discussions could be added in the manuscript. Proofread the paper and improve readability.Based on the above considerations, I recommend a minor revise.
Author Response
Reviewer 3
General Comments to Author:This paper conducted Hand-arm vibrations (HAV) assessment and monitored the changes in the thermal map of the skin in tennis athletes. In general, the paper is well written and easy-to-follow, with significant application prospect. The reviewer would suggest following few minor changes that could be made to improve the quality of the paper.
Specific Comments to Author:
Literature review could be more focused. Maybe there lacks detailed explanations about the key contributions. The readers need more help to understand what is important, what is new, and how it relates to the state of art. Some figures need improvement. Specifically, to my point of view, Figure 5 is not a good representative of the others. In “Results and discussion” section, authors give some figures. However, they do not make strong conclusions. What are the implications of the findings? More discussions could be added in the manuscript. Proofread the paper and improve readability.
Based on the above considerations, I recommend a minor revise.
Authors response: The structure of the paper was changed according suggesting of reviewers 1 and 2. The way as figure 5 is described changed, it now can be reads as: “Figure 5 shows one example of the vibratory signals obtained during the service for one of the various tests performed, which illustrates that the acquisition procedure was simultaneously time-dependent and direction-dependent.”
In authors point of view, the main implications of the paper are explained in conclusion but lets resume: “The IRT shows significant differences between the average skin temperatures for contralateral ROIs, which means that the risk of injuries is increased. The major differences are evaluated in the case of the extensor carpi radialis brevis, which could justify the frequent appearance of the injury known as tennis elbow. The results seem indicate also that athletes (numbers 5 and 6) with more experience and better positioned in the tennis ranking, show fewer differences in the contralateral skin temperature of some regions and smaller skin temperatures variation with exercise, when compared with the others. To reduce the adverse effect on the tennis athlete’s health some aspects should be considered for example, a suitable choice of racket, the tension of the string, considering using an anti-vibrator and developing a correct serve technique.”.
More discussions were added to the manuscript, discussions are now presented in the section 4.

Round 2
Reviewer 1 Report
Dear authors,
you have made a thorough study and the results are interesting. I think the paper has greatly improved by the changes you have made but you may still improve this paper so it becomes Clear what Your results really mean for the average Reader.
Firts the aim of this study is to evaluate HAV and changes in body temperature during the tennis serve.
Your hypothesis is that this will contribute to reduce risk of injury - this is not possible With Your design here. You may simply hypothesise that this will identify some risk factors that may be adressed by training (experienced tennis players vs novice, leg strength, grip) and by the Development of the racket.
You have a thorough description of how you did it and that is good, also the tables are more Reader friendly now.
The results are pretty Clear.
But:
The discussion needs to be improved and to be more stringent describing Your results in view of Your research question :
did the vibrations decrease? what made them decrease? what kind of influence did this have on skin temperature? what does this indicate?
are the results in line With previous research?
I suggest you cut out the lines 319-345 and move line 446- 464 to the start of the discussion and then Review Your conclusion in accordance.
what are the lessons are learned?
The Conclusion needs to be short and Sweet just summing up Your findings so please shorten and do not "discuss" here!
I also think you should have the text proof read before Publishing.
Author Response
Response to the reviewer’s comments:
The authors would like to thank the reviewers for reading the manuscript and for the comments and suggestions to improve the manuscript. Changes in the revised manuscript are highlighted with blue color.
Reviewer 1
Dear authors,
you have made a thorough study and the results are interesting. I think the paper has greatly improved by the changes you have made but you may still improve this paper so it becomes Clear what Your results really mean for the average Reader.
Authors response: we agree with the reviewer, sounds much better now.
Firts the aim of this study is to evaluate HAV and changes in body temperature during the tennis serve.
Your hypothesis is that this will contribute to reduce risk of injury - this is not possible With Your design here. You may simply hypothesise that this will identify some risk factors that may be adressed by training (experienced tennis players vs novice, leg strength, grip) and by the Development of the racket.
Authors response: we suppose that reviewer is referred to abstract and, therefore, we change the abstract and it can now be read as:
“During recent years the number of tennis athletes has increased significantly. When playing tennis, the human body is exposed to many situations which can lead to human injuries, such as the so-called tennis elbow (lateral epicondylitis). In this work a biomechanical analysis of the tennis athletes, particularly during the service, was performed, considering three different types of over-grip and the presence of one anti-vibrator device.
One part of the study evaluates the exposure to hand-arm vibration of the athlete, based on the European Directive 2002/44/EC concerning the minimum health and safety requirements, regarding the exposure of workers to risks from physical agents. The second part of the study considers an infrared thermography analysis in order to identify signs of risk of injury, particularly the tennis elbow, one of the most common injuries in this sport.
The results show that the presence of the anti-vibrator influences the vibration values greatly. The presence of the Cork and/or Tourna on the racket grip does not have any significant effect on the HAV and, in fact, the infrared thermography technique can be used to identify the risk of injuries in tennis players.”
You have a thorough description of how you did it and that is good, also the tables are more Reader friendly now.
Authors response: thanks.
The results are pretty Clear. But:
The discussion needs to be improved and to be more stringent describing Your results in view of Your research question:
Authors response: author made some adjustment in the discussion in order to remove some text related with the description of some published work.
We removed the text in lines 369-374, however, the references were maintained.
We removed the text in lines 399-403, however, the references were maintained.
we change the test within lines 405-427.
We removed the lines 441-442, however, the reference was maintained
did the vibrations decrease?
Authors response: in the discussion is possible to see that authors made a clear statement related with the presence of anti-vibrator: “In fact, from the results present in Tables 4 and 5 it is possible to conclude that the presence of the anti-vibrator reduces the HAV by around 2% to 7%.” In the conclusion section this idea is also present in the first paragraph.
Moreover, it was highlighted in yellow the following text: “The cork over-grip should, theoretically, reduce vibrations due to the material characteristics of cork. However, the HAV values increase by around 6%, which can be explained by the difficulty the athletes had in holding the racket firmly, owing to the greater thickness of this over-grip, which means that the athlete must use greater force to service, which increases the vibrations induced. The Tourna grip, when compared with the standard Wilson Pro, does not present significant differences.” In the conclusion it can now be read as: “The presence of Cork and/or Tourna on the racket grip does not have any significant effect on the HAV. The values obtained for the HAV are very high and can’t contribute to diminish the risk of injuries for the tennis athlete. In the case of HAV exposure, the daily level of the total magnitude of vibration exceeded the limit of the daily exposure action value, in all situations, which indicates the need for a rapid intervention.”
Furthermore, the statement of lines 357-364, can now read as: “Statistical analysis of the results presented through Tables 4-7 have shown that the presence of the anti-vibrator and Tourna or Cork over-grips can affect athletes differently: for athlete 1, the presence of cork, when compared to Wilson Pro racket with anti-vibrator, affects the values of RMS in the X and Y directions, with p values of 0.043 and 0.0296, respectively; For athlete 2, the RMS Y values of the pair Wilson pro racket with and without anti-vibrator show statistically significant differences, with p equal to 0.017,.and the presence of cork on the grip has also statistical significance differences in all the values of RMS, when compared to WGWAV; in the case of athletes 3 and 4 no statistically significance differences are obtained (p>0.05); for the athletes 5 and 6, the TGWAV changes the HAV values significantly, when compared to the WGWAV, for all the RMS values, with p values well below 0.05”
what made them decrease?
Authors response: From the answer to the previous question it is clear that the anti-vibrator device was the main factor that contribute to this decrease.
what kind of influence did this have on skin temperature?
Authors response: Actually, the results of the IRT technology were obtained for the racket Wilson pro with anti-vibrator. So, because we not perform similar measurements for the other two grip conditions, we can’t compare the thermal effect among different grip types. Nevertheless, it would be expected similar skin temperature variations. In order to clarify this issue, the following text was placed after figure 4:
“The racket used was the Wilson Pro with anti-vibrator”.
what does this indicate?
Authors response: The IRT technology can be used to identify the regions that are more solicited and, of course, is one of the most often used indicators of health status. So, in order to give some continuity on the discussion, we change the text within lines 405-427 it can now be read as:
“The infrared thermography technology can be used to identify the regions that are more solicited during physical activity and exercise, and is one of the most often used indicators of health status in humans [49]. From the results presented in Table 8, it may be seen that for almost all the athletes and for all the muscles under analysis, there is a decrease in the skin temperature, which agree with previous works [25,49,50], and can be explained by the cutaneous vasoconstrictor response to exercise. In fact, the results of Table 8 show that for some athletes the differences are higher than 0.5ºC and it may require attention. Moreover, from the results of the contralateral skin temperature for the different regions of interest presented in Table 9, is possible to see that major variances occur in the RECRB/LECRB contralateral ROIs, followed by the RB/LB contralateral regions, with values that are higher than 0.4ºC in four athletes. These observations may justify the frequent appearance of the injury known as tennis elbow. In fact, Hildebrandt et al. [51] says that a difference of more than 1ºC between contralateral ROIs can indicate a pathophysiological process and Marins et al. [32] concludes that a difference higher than 1.6ºC for the contralateral ROIs can be considered “High Severity”, in the level of attention, and a difference less than 0.5ºC no special attention is required. Regarding the p values presented in Table 9, it can be concluded that p is smaller than 0.05 for the comparison between RECRB/LECRB, for all the athletes, and for three athletes in the case of RB/LB. So, this means that there is a statistically significant difference between the two contralateral ROIs under analysis, especially the extensor carpi radialis brevis, which confirm the possibility appearance of the tennis elbow injury.”
are the results in line with previous research?
Authors response: From the references placed within the discussion section, it is possible to conclude that the behavior of the skin temperature variations is similar to that reported in some published work. Nevertheless, to the best of the authors’ knowledge, there is no published work that relates the IRT with tennis players.
I suggest you cut out the lines 319-345 and move line 446- 464 to the start of the discussion and then. Review Your conclusion in accordance.
Authors response: the paper was modified, according reviewer suggestions. Moreover, we changed the position of the text placed between lines 365-389, to just after Line 356. These line numbers are from the older version of the revised paper. We also change the position of lines 357-364 to after the discussion of the results related with all tables.
what are the lessons are learned?
Authors response: it was confirmed that the presence of the anti-vibrator influences the vibration values greatly. The presence of the Cork and/or Tourna on the racket grip does not have any significant effect on the HAV. The IRT technique can be used to identify the risk of injuries in tennis players.
The Conclusion needs to be short and Sweet just summing up Your findings so please shorten and do not "discuss" here!
Authors response: it can now read as:
” In terms of the hand-arm vibration induced, it is possible to conclude that the best situation, in terms of the athlete’s health, with less vibration transmitted, is the Wilson Pro racket with anti-vibrator (WGWAV), and the worst case is the WGNAV. So, it can be confirmed that the presence of the anti-vibrator influences the vibration values greatly. The presence of Cork and/or Tourna on the racket grip does not have any significant effect on the HAV. The values obtained for the HAV are very high and can’t contribute to diminish the risk of injuries for the tennis athlete. In the case of HAV exposure, the daily level of the total magnitude of vibration exceeded the limit of the daily exposure action value, in all situations, which indicates the need for a rapid intervention. The IRT shows significant differences between the average skin temperatures for contralateral ROIs, which means that this technique can be used to identify the risk of injuries in tennis players.”
I also think you should have the text proof read before Publishing.
Authors response: we proceed as suggested by reviewer.

Reviewer 2 Report
Thank you for submitting a revised version of the manuscript: “Hand-arm vibrations assessment and changes in the thermal map of the skin in tennis athletes during the service”, ijerph-647229.
I think that the paper was improved since the previous submission. And it can be accepted with some minor comments:
Line 329: I suggest to remove this sentence.
Line 332: I suggest to remove or change this sentence.
Line 335: Please rephrase.
Figure 5 is horrible. Please consider to re-draw the figure with appropriate software. From the updated caption I understand that the unit of measurements for the vertical axis is (m/s2). Wouldn’t it be better to express it in (g) ?
Author Response
Response to the reviewer’s comments:
The authors would like to thank the reviewers for reading the manuscript and for the comments and suggestions to improve the manuscript. Changes in the revised manuscript are highlighted with blue color.
Reviewer 2
Thank you for submitting a revised version of the manuscript: “Hand-arm vibrations assessment and changes in the thermal map of the skin in tennis athletes during the service”, ijerph-647229.
I think that the paper was improved since the previous submission. And it can be accepted with some minor comments:
Authors response: we agree with the reviewer, sounds much better now.
Line 329: I suggest to remove this sentence.
Authors response: the sentence was removed. Actually, it can be seen that we have removed the lines 319-345, as suggested by Reviewer 1.
Line 332: I suggest to remove or change this sentence.
Authors response: the sentence was removed. Actually, it can be seen that we have removed the lines 319-345, as suggested by Reviewer 1.
Line 335: Please rephrase.
Authors response: the sentence was removed. Actually, it can be seen that we have removed the lines 319-345, as suggested by Reviewer 1.
Figure 5 is horrible. Please consider to re-draw the figure with appropriate software. From the updated caption I understand that the unit of measurements for the vertical axis is (m/s2). Wouldn’t it be better to express it in (g) ?
Authors response: we decide to remove the figure and, therefore, we removed the following text: “Figure 5 shows one example of the vibratory signals obtained during the service for one of the various tests performed, which illustrates that the acquisition procedure was simultaneously time-dependent and direction-dependent.” And figures were renumbered according.

Round 3
Reviewer 1 Report
Abstract: l 20 varies greatly?
l 21 does not have any significant effect on hand-arm vibration (HAV).
cut out "in fact" and start the sentence : The results indicate that infra-red thermography MAY be used..."
Conclusion:
p 16 l 421 "health" is perhaps a strong word since it is a more local injury in the arm and hand...
Author Response
Response to the reviewer’s comments:
The authors would like to thank the reviewer for reading the manuscript and for the comments and suggestions to improve the manuscript. Changes in the revised manuscript are highlighted with blue color.
Reviewer 1
Abstract: l 20 varies greatly?
Authors response: The abstract was changed according to the reviewer suggestion: “values greatly in the case of the athlete with more experience and also for the athlete with less performance.”
l 21 does not have any significant effect on hand-arm vibration (HAV).
Authors response: The abstract was changed according to the reviewer suggestion: “similarly in the case of the athlete with the best performance and the athlete with less technique.”
cut out "in fact" and start the sentence : The results indicate that infra-red thermography MAY be used..."
Authors response: The sentence was changed according to the reviewer suggestion: “The results indicated that the infrared thermography technique may be…”
Conclusion:
p 16 l 421 "health" is perhaps a strong word since it is a more local injury in the arm and hand...
Authors response: The conclusion was changed according to the reviewer suggestion: “to avoid injuries to athlete's arm, …”
